# Advanced Heat Treatment of Pearlitic Rail Steel

**DOI:** 10.3390/ma16196430

**Published:** 2023-09-27

**Authors:** Magdalena Jabłońska, Filip Lewandowski, Bartosz Chmiela, Zbigniew Gronostajski

**Affiliations:** 1Faculty of Materials Engineering, Silesian University of Technology, Krasińskiego 8, 40-019 Katowice, Poland; bartosz.chmiela@polsl.pl; 2Faculty of Mechanical Engineering, Wrocław University of Technology, Ignacego Łukasiewicza 5, 50-371 Wrocław, Poland; filip.lewandowski@pwr.edu.pl

**Keywords:** heat treatment, rail profiles, pearlitic steel

## Abstract

The aim of this research is a systematic investigation of heat treatments of 60E1 profile rails made of steel R350HT, which would ensure the properties required by the standard EN 16273. Additionally, it presents a concept of cooling rails on a semi-industrial station, which will make it possible to obtain the desired properties. The dilatometric tests have demonstrated that the optimal cooling rate is within the scope of 3 °C/s to 6 °C/s, when both the EN 16273 standard’s hardness distribution and microstructure requirements are fulfilled. The tests on the designed and built station showed that the optimal pressure with respect to the microstructure and properties of the rail equals 6.5 bar. For these parameters, measurements of the interlamellar distance were also performed—the cooling rate obtained at the surface was 3.68 °C/s, with an interlamellar distance of about 80 nm, whereas inside the rail the rate was 2.63 °C/s and the distance 110 nm. The achieved results confirm that the designed station can be used for controlled cooling of rail steels from R350HT steel.

## 1. Introduction

Studies referring to the properties of pearlitic steels used in railway engineering have already been conducted in the second half of the 20th century [1]. In time, together with the technical progress and the general need of building new railway lines according to the technical interoperability standards [2], a necessity occurred to standardize the steel grades used to make the railways as well as to determine standards for their production [3]. It should be mentioned that there is also research into bainitic steel for application in rails [4]. The European Standard for railway rails is given in two norms [5,6].

At present, the most popular grades of pearlitic steels used for rails are R260 and R350HT in the European Union [5], and U75V in Middle Eastern countries [7]. The chemical compositions of these steels are shown in Table 1. They are very similar, but rails made of steel R350HT additionally undergo thermal treatment and are characterized by better properties, thus finding more and more applications. In the past, most rails were made of R260 steel, and they are still tested for, among other things, fatigue [8] and white layer [9]. Recently, R350HT has been used more and more often, but the heat treatment process is complicated in the case of large-scale production. Many metallurgical companies try to optimize this technology for final properties and performance, but the literature provides very little information on the heat treatment of R350HT steel.

In order to obtain the desired strength properties of railway rails, thermal treatment procedures are applied to the rail’s head [10]. The basic properties of railway rails made of steel R350HT required by the standards [5] as well as the operation conditions are [11]:Fine inter-lamellar spacing between the cementite lamellae in the pearlite;Lack of a bainitic structure:High hardness (HV > 350);Lack of a rapid hardness drop from the head’s surface progressing towards the inside of the rail’s cross-section;Lack of ferritic precipitates.

The literature has frequently made attempts at determining the dependence between the microstructure morphology [12], and the properties of steels [13]. Based on the strength tests conducted by means of X-ray, a synchrotron [14] and neutron diffraction [15] it was established that, for a structure consisting of 100% pearlite, the steel properties depend on the cementite shape, the grain size and, most of all, the distance between the cementite lamellae in the pearlite (Sp).

The study [16] presents the effect of the morphology of pearlite microstructure on the strength and workability of pearlitic steel C70. It establishes that, for a given volume fraction of cementite, the strength of pearlitic steel is mainly determined by the distance between the cementite lamellae in the pearlite. The research was conducted for two fully pearlitic structures, with distances between the cementite lamellae of 230 and 170 nm, for which yield points of 396 MPa and 498 MPa and a hardness of 220 HV50 and 270 HV50, respectively, were obtained.

The paper [17] shows an analysis of the effect of the cementite’s lamellae distance in pearlite (Sp) on the hardening coefficient. It was demonstrated that, in the scope of the interlamellar distance values of 140–280 nm, the hardening effect is larger than above this value, where the hardening significantly decreases. Based on that, a dependence was formulated between the cementite’s lamellae distance in pearlite and the hardening coefficient, in the form of
(1)$$n=-0.52\cdot \exp\left(\frac{-\mathrm{Sp}}{58.29}\right)+0.32$$

There are also numerous studies whose authors attempt to describe the yield stress in pearlitic steels with the Hall–Petch relation, by replacing the mean grain size with the distance between the cementite lamellae in pearlite, such as the works of Sevillano [18] and Ray and Mondal [19].

In the study [20], the mechanical properties of steel with an almost fully pearlitic structure, with a carbon content of 0.65% C, were described in the function of the interlamellar distances. The steel was thermally treated at different austenitization temperatures in order to obtain different interlamellar distances. It was observed that the hardness and the yield point were in agreement with the Hall–Petch relation in reference to the distances between the cementite lamellae in the pearlite, whereas for the tensile strength (UTS) the percentage of elongation was not described by the Hall–Petch relation. It was noticed that, below the critical value of this distance, the UTS, the impact strength and the plasticity do not undergo a significant change. However, in these investigations, the steel had quite a big distance between the cementite lamellae in the pearlite—at a level of over 500 nm.

The work [21] presents predictions of pearlitic steel hardening during cold deformation by means of a phenomenological model and Taylor’s micro-mechanical model. The phenomenological model is based on Bouaziz’s model.

Research is also carried out on the influence of welding parameters on the microstructure of pearlite. In the manuscript [22], the authors present the influence of thermit welding on the mechanical properties of the rails. The continuation of this research is an analysis of the microstructure which proves that after welding fine pearlite grains and large coarse pearlite grains are formed as a function of the cooling rate [23].

In the case of the thermal treatment of railway steels, there is a lot of information dating as far back as to the 1970s, with suggestions for methods for their cooling. Decreasing the distance between the cementite lamellae in pearlite causes an increase in strength and hardness of the pearlitic structure, which is directly connected with a better wear resistance of the rail head’s pitch surface. The literature provides descriptions of methods of accelerated cooling with the use of water [24], water–air mist [25], compressed air [26] and by way of submersion in water–polymer solutions [8]. Most of the manuscripts present general solutions without details, for example on the correlation between cementite’s lamella distance in pearlite and cooling rate and mechanical properties. The literature certainly lacks such information, especially for steel R350HT.

In the case of steel R350 HT, despite its very great popularity due to the application for railway rails, the literature does not include a lot of studies on its microstructure, especially with respect to the connection between the mechanical properties of this steel and the distance between the cementite lamellae in the pearlite. A certain exception is the research conducted by prof. Kuziak’s team, who not only undertook detailed investigations of the microstructure but also performed modelling of it. The work [27] presents a mathematical model based on the metal science knowledge which makes it possible to design the parameters of the thermal treatment of a rail head made of pearlitic steel in order to obtain the required mechanical properties. Owing to the implementation of the model in a computer program based on the finite element method, it is possible to track the progress of the phase transformations during the thermal treatment of a rail head with a pearlitic structure and, in the end, to predict the parameters of the final microstructure as well as the mechanical properties of the rail. The scope of the examined cooling rates was 0.25 °C/s, 1 °C/s, 5 °C/s and 10 °C/s. An expansion of these studies is the work [28], which presents a new approach to the optimization of the thermal treatment of pearlitic rails made of steel R260 aimed at obtaining an advantageous relation between hardness and ductility of the rail head’s pitch layer.

Our analysis of the literature has shown that there are a lot of research studies referring to steel R260, most of which focus on raw materials not taken from rolled rails. The literature provides very little information on the thermal treatment of steel R350HT. There are attempts at relating the cementite’s lamella distance in pearlite to the mechanical properties, yet there are no data referring to a relation between these properties and the cooling rates, which would facilitate the design of the thermal treatment technology. The aim of this research is a systematic investigation of thermal treatments of 60E1 profile rails made of steel R350HT, which would ensure the properties required by the standard EN 16273 [5]. Additionally, it presents a concept of cooling rails on a semi-industrial station, which will make it possible to obtain the desired properties.

## 2. Methodology

In order to determine the optimal thermal treatment concept, the following investigations were carried out:Determination of the effect of the cooling rate on the microstructure and hardness of steel R350HT in dilatometric tests;Construction of a station simulating the real rail cooling process under semi-industrial conditions;Determination of the relation between the cooling parameters of rails made of steel R350HT and its microstructure and properties on the constructed station.

According to the standard EN 16273 [5] the hardness along the line from T1 to T4 (Figure 1a) in the rail’s head after the thermal treatment should be within the acceptable area marked in Figure 1b and the microstructure should not contain bainite or a closed ferrite network.

The course of the thermoplastic treatment during the production of railway rails from steel R350HT is shown in Figure 2. This study refers to the part of the heat treatment marked in green.

For a more precise description of the microstructure morphology, characteristic features should be properly defined. In a measurement of the pearlite’s interlamellar distance, it is important to consider the fact of a uniform distribution of pearlite in the whole material. For measurements, it is necessary to analyze the inter-lamellar distances for the areas appropriate in terms of the inclination of the pearlite lamellae to the observation plane (Figure 3).

The literature includes a lot of methods for measuring the distance between the cementite lamellae in pearlite, two of which seem the most comfortable. The first one is the CLM (circular line method) [18], which consists in containing the boundary of three pearlite colonies within a circle of a given diameter and determining the number of intersections of the cementite lamellae with the circle. This method is illustrated in Figure 4a (Equation (1)). This method can be used for such places in the microstructure where, at the junction of three grains, we are dealing with a parallel arrangement of pearlite lamellae. The other method is the LIM (linear intercept method) [22], which consists in determining a section of a given length within the area of a pearlite colony and then determining the number of intersections of the cementite lamellae with this section. An example of this method is shown in Figure 4b. In the study, the first method was selected (CLM), where, for each set of cooling conditions, measurements of five microsections were made, followed by the calculation of the mean value.
Sp = 1/2 × L/N,(2)
where Sp—interlamellar distance, L—circle circumference and N—number of intersections of the lamellae with the circle for a given sample.

## 3. The Effect of the Cooling Rate on the Microstructure and Hardness of Steel R350HT in Dilatometric Tests

In the process of thermal treatment of pearlitic steels, the key parameter is the cooling rate of the material. The cooling rate influences the morphology of the formation of a pearlite colony. In order to obtain the expected steel properties in agreement with the standards, it is necessary to determine the desired thermal treatment parameters. The hardness of rail steels after thermal treatment should be within the scope of 300–400 HV. Such hardness in the rail steel used on the European and Asian market (differing only in the name and, slightly, the chemical composition) has been obtained for the interlamellar distance within the scope of 80–125 nm. Unfortunately, these tests were conducted on samples from wrought material (not from rolled rails) or on the basis of numerical simulations [16,27].

In order to determine such a relation for rolled samples after plastic deformation and verify the correctness of the results provided in the literature as well as search for the proper thermal treatment conditions, a decision was made to perform investigations of the effect of the cooling rate on the temperature of the occurring pearlitic transformation, and the hardness and the microstructure of steel R350HT on samples cut out of the rails. Dilatometric tests were conducted, which were based on monitoring the changes in the volume of the phases present in metal alloys, on the basis of the dependence of the sample length on the cooling temperature and rate. The investigations were carried out on a Dilatometer DIL805A/D/T produced by TA Instruments, equipped with a measuring head type LVDT, with a theoretical resolution of +/−0.05 mm, as well as a laser with a resolution of +/−0.1 mm. The dilatometer has a computer system for controlling and recording the experimental data. The samples used in the tests were collected from the upper fragment of the rail head made of grade R350HT and had the shape of cylinder with a ring cross-section, where the external diameter was φ = 4 mm and the internal diameter equaled φ = 2 mm. The heating of the sample was realized by means of the induction method, with the use of a generator with the frequency of 250 kHz. It was performed in the atmosphere of argon. The temperature deviations did not exceed +/−0.3°. The cooling was carried out with compressed nitrogen.

Based on the presented state of the problem and on the basis of a CTPc diagram (Figure 5), which was determined using program JMatPro v. 7.0, the following thermal treatment conditions were selected: austenitization temperature, 900 °C; austenitization time, 150 s. The time of linear heating to the austenitization temperature equaled 240 s. The most important parameter of the thermal treatment of pearlitic steels, including rail steels, is the cooling rate during the formation of a pearlite colony, and so a wide scope of cooling rates was planned, including the following rates 10 °C/s, 8 °C/s, 6 °C/s, 4 °C/s, 3 °C/s and 2 °C/s (Figure 6).

The result of the performed trials in the form of dilatometers is presented in Figure 7. The determined temperatures of the transformation’s beginning and end are included in Table 2. The method of determining the transformation’s beginning and end temperatures itself was based on the use of a graphical method, which consists in reading out the temperature value in the points where the linear course of the cooling curve ends (in the case of the transformation beginning) or begins (in the case of the transformation ending). The ending or beginning point of the linear course of the cooling curve is considered to be the point where the curve in the transformation area intersects with the tangent.

Together with the increase in the cooling rate, the pearlitic transformation’s beginning and end temperatures become lower. For the highest assumed cooling rates (10 °C/s and 8 °C/s), the dilatation effect from the pearlitic transformation overlaps with the dilatation effect in the scope characteristic for the bainitic transformation. This could suggest the presence of a phase different than pearlite in the microstructure of the thermally treated samples, and so, for these samples, microstructural tests were conducted, which confirmed the formation of a bainitic structure under these conditions as well as the fact that the cooling rates 10 °C/s and 8 °C/s are too high to obtain a pearlitic structure in the whole volume (Figure 8).

The following stage of the conducted research was the determination of the interlamellar distance in the samples in which a pearlitic structure had been obtained. The measurement of the mean interlamellar distance was made by means of the CLM method (circular line method). For each cooling rate, the measurements were made for each randomly selected microsection fragment. From all the collected results, the arithmetic mean was calculated, which describes the mean distance between the cementite lamellae in the pearlite for the given cooling rate.

It should be mentioned that the CLM method is not justified for all the performed trials. For the cooling rates of 10 °C/s and 8 °C/s, the presence of bainite and of so-called degenerate pearlite in the material’s structure was observed. Figure 9a presents a photograph of the pearlitic structure, whereas Figure 9b shows the bainitic structure with degenerate pearlite, for which neither the CLM method nor any other method makes it possible to properly determine the distances between the cementite lamellae in the pearlite. Figure 10 presents the microstructures for the other cooling rates.

Exemplary results for a cooling rate of 2 °C/s for five different microsections are shown in Table 3, where the mean value is 115.1 mm.

A diagram of the dependence of the interlamellar distance on the cooling rate is presented in Figure 11. We should note the fact that, for the cooling rate increasing above 3 °C/s, the obtained interlamellar distance values were very similar—at a level of Sp = 80 +/− 3 nm. This means that there is a certain critical value of the cooling rate above which the interlamellar distance value does not decrease. In turn, for higher cooling rates, i.e., 10 °C/s and 8 °C/s, a bainitic structure was obtained, which means that the character of the transformation changes from a diffusive one into a partially diffusive one.

In the following stage the relation between the cooling rate and the material’s hardness was determined. It is presented in Figure 12.

It is noticeable that the hardness obtained for a cooling rate of 2 °C/s does not fulfill the requirements for the whole hardness scope in rails made of steel R350HT (Figure 1). In turn, for the cooling rates of 8 °C/s and 10 °C/s the obtained hardness values on the sample surfaces were too high and exceeded the acceptable scope. The hardness at a level of 345 HV obtained for 2 °C/s is acceptable already for the measurement made at a distance of 4 mm from the head surface of the rail. And so, at certain distances from the surface such a cooling rate should be considered acceptable.

Increasing the cooling rate to above 3 °C/s does not significantly affect the distances between the cementite lamellae but causes an increase in the hardness. Then, slight changes in the distance between the cementite lamellae cause significant increases in the hardness, which are probably mainly caused by the cooling rate and the decrease in the carbon’s diffusion rate as well as the increase in the amount of carbon in the ferrite (Figure 13).

## 4. Simulating the Real Process of Cooling Rails under Semi-Industrial Conditions

After the determination of the cooling rates ensuring the proper microstructure and properties of steel R350HT, a concept of a cooling station was developed, which would make it possible to control a full rail.

The main assumptions for the station are:A rigid construction, ensuring a constant position of the cooling agent’s feeder with respect to the rail;Resistance to high temperatures (<1100 °C);A possibility of arranging the nozzles supplying the cooling agent in the optimal way, ensuring a uniform distribution of the cooling rate on the cross-section of the rail head;The station should constitute the starting point for a fully industrialized line for the thermal treatment of rails.

On the basis of the dilatometric tests, it was established that the desired cooling rate should be within a scope of 1–6 °C/s. Such a cooling rate scope makes it possible to achieve a higher hardness in the upper areas of the cross-sections and a respectively lower hardness in the lower areas.

In reference to the set assumptions and the available visualizations of the potential solutions of this type, a solution was designed in the form of a casing with the shape of an even-armed trapezoid, on which, around the rail head’s outline, six identical air nozzles were mounted symmetrically, three on each side, directed successively towards the side surface, the rounded head surface and the pitch surface. The design assuming the casing’s dimensions and the distance between the air nozzle fronts are shown in Figure 14. Based on the preliminary tests conducted on rail head fragments, the proper cooling medium was determined to be compressed air, which, depending on the pressure, made it possible to steer the cooling rate within a wide range. Both its working pressure and the pressure supplied directly to the device were regulated, within a scope of 1–7 bar. The construction of the installation included the use of hydraulic hoses as well as elements separating and limiting the flow of the supplied air. Figure 14 shows a diagram of the construction of the finished station, whereas Figure 15 presents the cooling tests. The presented cooling station has been submitted for patent protection.

The conduits exiting the dividing strip were not connected to the successive push fits but were connected in pairs with respect to the symmetry axis. That is, the nozzles cooling the pitch surface were connected to opening 1 and 2, the rounded part of the head to opening 3 and 4, and the pitch surface to opening 5 and 6. Such a procedure aimed at a possibly uniform distribution of the cooling agent’s flow on both halves of the cross-section of the thermally processed rail.

Figure 16 presents the concept of a fully industrial station for the thermal treatment of rails made of steel R350 HT with the profile 60E1 based on the proposed station.

In order to verify the proper operation of the constructed station, thermal treatment trials were conducted on 100 mm long rails made of steel R350HT with the profile 60E1. The measurement of the cooling rate in the rail was made at points 1, 2, 3 and 4 marked in Figure 17, in which holes were drilled for K type thermocouples, with a diameter of φ = 2 mm.

The cooling temperature was measured with TandD’s MCR-4TC, where all thermocouples were mounted at the same time. The sampling rate was 1 Hz, data were collected during the course of the entire test (during heating and cooling). The cooling rate was determined using a graphical method on the basis of the curve from start of accelerated cooling to pearlitic transformation. The accuracy of the K-type thermocouple was (±0.5 °C + 0.3% of reading) and the operating range for the set was −270 to 1370 °C.

In order to determine the effect of the working pressure on the cooling rate in the particular areas of the rail, measurements were made of the cooling rate for the compressed air working pressures of 0.5 to 6.5 bar. Exemplary cooling rate distributions for the working pressures of 0.5 bar and 6.5 bar are presented in Figure 18. Table 4 compiles the cooling rate results for all the applied pressures. The lowest cooling rate in the most experiments was at the point 4; however, cooling rates at point 3 and 2 were very similar. Therefore, the interlamellar distance for 6.5 bar was determined only for points 1 and 3 and was about 80 nm and 110 nm, respectively.

Based on the dilatometric tests, the hardness measurements and the standard [5] a decision was made to apply a compressed air pressure of 6.5 bar during the thermal treatment. In the following, for samples cooled at a rate of 6.5 bar, hardness and microstructure tests were conducted. The measurement points were set according to the standard EN 16273 (Figure 1).

On the thermally processed samples, a hardness measurement was made at 60 points—15 each, arranged along lines T1–T4, where the first was located 1 mm under the surface of the rail and each following one 2 mm further. The localization of the tests and the acceptable hardness scope are shown in Figure 19.

Based on the measured cooling rates, it was possible to conclude that the constructed station worked acceptably, and the cooling rate obtained during the tests was within the scope determined during the dilatometric tests. What is more, the achieved hardness also fulfilled the conditions of the considered standards. The hardness rail from R350HT steel without heat treatment equals about 320 HV30 (300HB) in all the cross-sections, which is much lower than the requirements.

For the obtained microstructures, measurements were also made of the interlamellar distance for a pressure of 6.5 bar according to the method described above, for the following measurement points: at point 1 with a cooling rate of 3.68 °C/s and at point 3 with one of 2.63 °C/s. The obtained interlamellar distances were about 80 nm and 110 nm, respectively (Figure 20). The achieved results confirm that the designed station can be used for trials of controlled cooling of rail steels.

## 5. Conclusions

The dilatometric tests have demonstrated that the optimal cooling rate is within a scope of 3 °C/s to 6 °C/s, for both the EN 16273 standard’s hardness distribution and microstructure requirements to be fulfilled. For a cooling rate of 2 °C/s, the required hardness is reached only a few millimeters from the material’s external surface, and for the cooling rates of 8 °C/s and 10 °C/s, the requirements are not met, not only due to too high hardness but also the presence of a bainitic structure, undesirable according to the standards, as well as degenerate pearlite.

On the basis of the obtained results, we can state that together with an increase in the cooling rate within the scope of 2 °C/s to 6 °C/s, the material’s hardness increases and the distance between the cementite lamellae in the pearlite decreases (S_p_). However, above a rate of 3 °C/s this distance decreases much slower. The scope of the cementite’s lamella distance for the microstructures meeting the hardness distribution requirements was within a scope of 77–115 nm.

The study has also developed a concept and construction of a cooling station, which will make it possible to perform controlled cooling of a full rail. The proper cooling medium was established to be compressed air, which, depending on the pressure, made it possible to steer the cooling rate in a wide range. Both its working pressure and the pressure supplied directly to the device were regulated within a scope of 1–7 bar. The tests showed that the optimal pressure, with respect to the microstructure and properties of the rail, equals 6.5 bar. For these parameters, measurements of the interlamellar distance were also made—the cooling rate obtained at the surface was 3.68 °C/s, with an interlamellar distance of about 80 nm, whereas inside the rail, the values were 2.63 °C/s and 110 nm. The achieved results confirm that the designed station can be used for tests of controlled cooling of rail steels.

## Figures and Tables

**Figure 1 materials-16-06430-f001:**
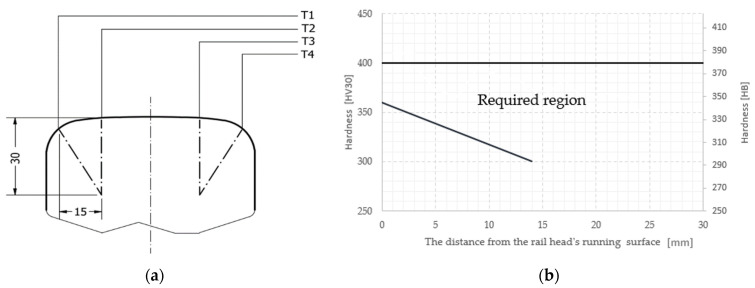
Requirements of the standard EN 16273 referring to (**a**) the location of the hardness measurement and (**b**) the scope of acceptable hardness depending on the distance from the rail head’s front surface [5].

**Figure 2 materials-16-06430-f002:**
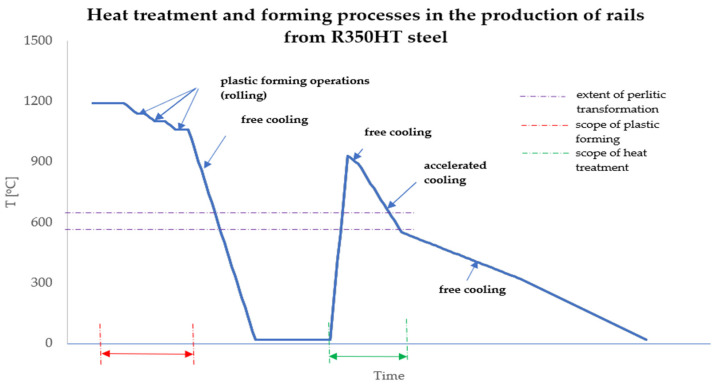
Diagram of the thermomechanical treatment of rails made of steel R350HT.

**Figure 3 materials-16-06430-f003:**
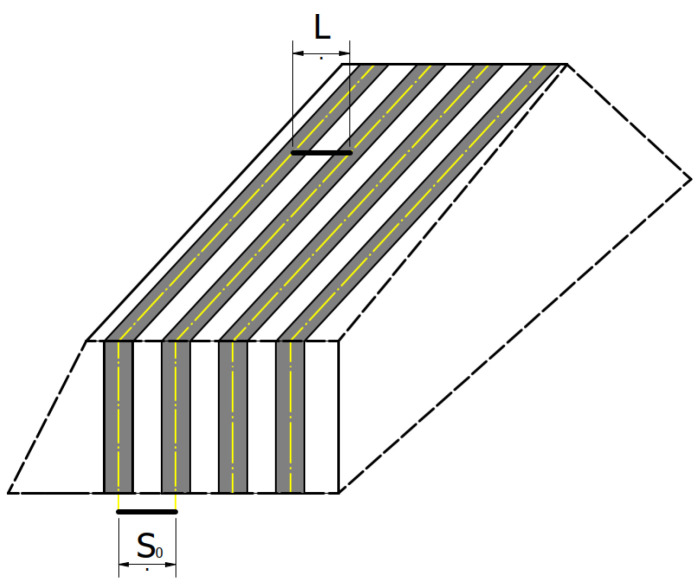
Schematic presentation of the measure real distance the pearlite lamellae (own design). S_0_—real distance between the pearlite lamellae. L—line segment between the two axes of the cementite lamellae.

**Figure 4 materials-16-06430-f004:**
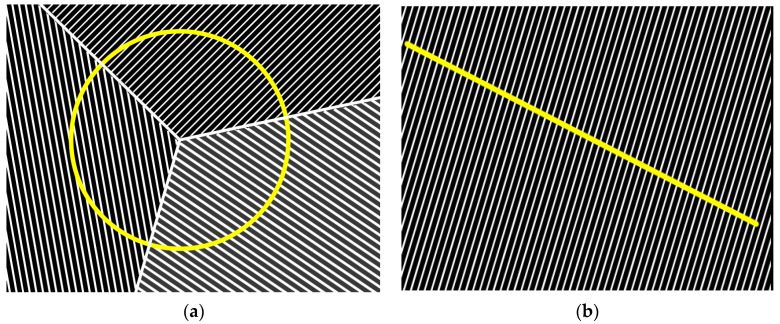
Schematically presented method of measuring the cementite’s lamellae distance in pearlite: (**a**) CLM [19] and (**b**) LIM [15]. Determiantion number of intersections of the cementite lamellae with yellow circle and line.

**Figure 5 materials-16-06430-f005:**
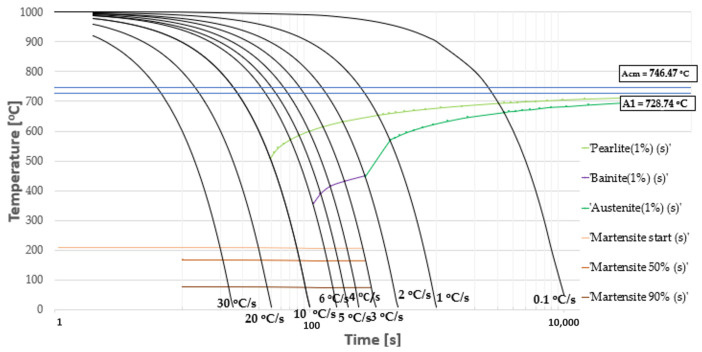
CTPc diagram for the investigated steel with a eutectoid composition.

**Figure 6 materials-16-06430-f006:**
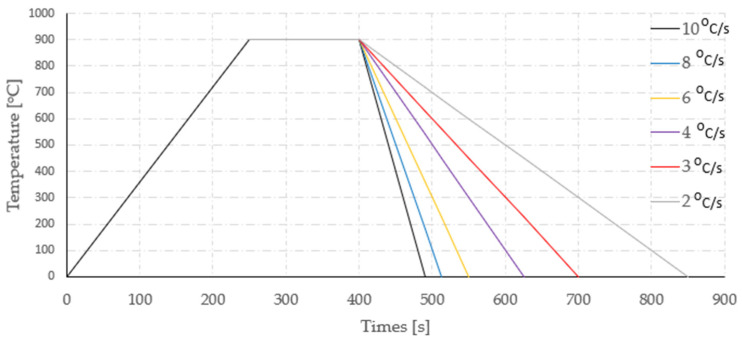
Course of temperature in the function of time for the tests.

**Figure 7 materials-16-06430-f007:**
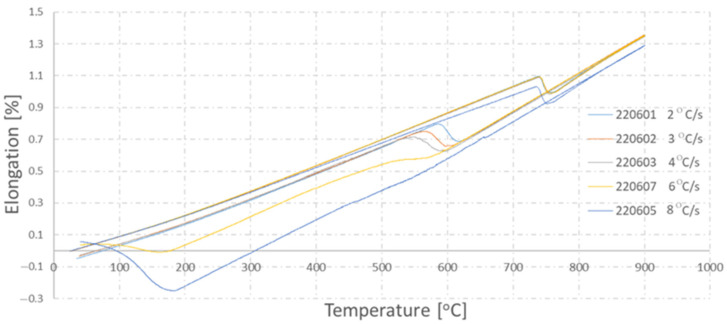
Dilatometric curves for tested steel.

**Figure 8 materials-16-06430-f008:**
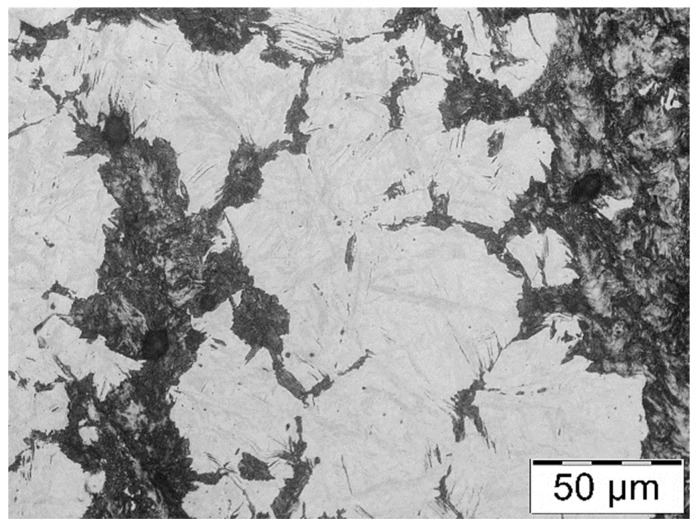
Microstructure of steel cooled at a rate of 8 °C/s with visible areas where a bainitic structure was formed.

**Figure 9 materials-16-06430-f009:**
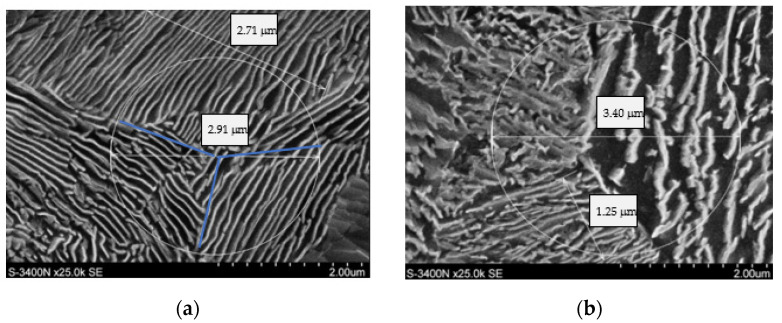
Microstructure of pearlite for the cooling rates (**a**) 6 °C/s and (**b**) 8 °C/s.

**Figure 10 materials-16-06430-f010:**
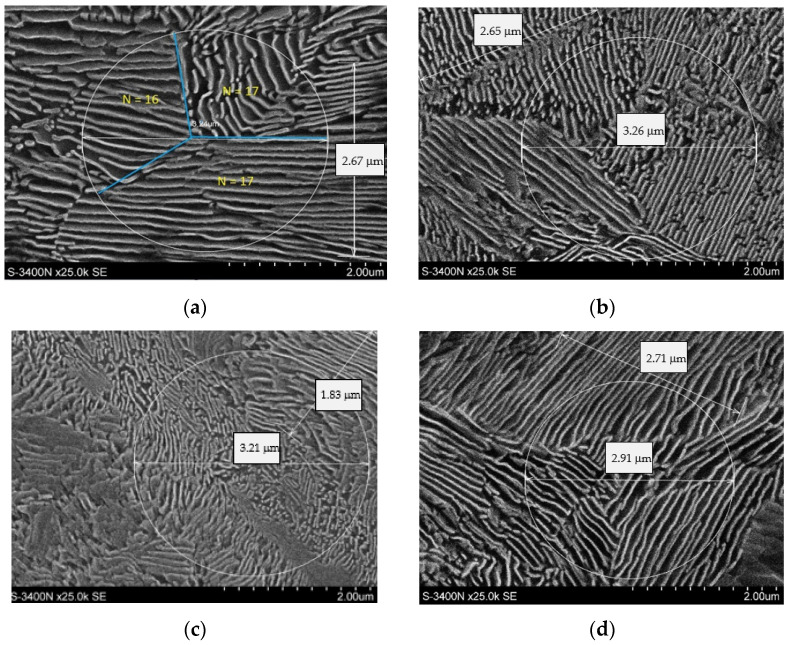
Microstructures of perlite for the cooling rates (**a**) 2 °C/s, (**b**) 3 °C/s, (**c**) 4 °C/s and (**d**) 6 °C/s.

**Figure 11 materials-16-06430-f011:**
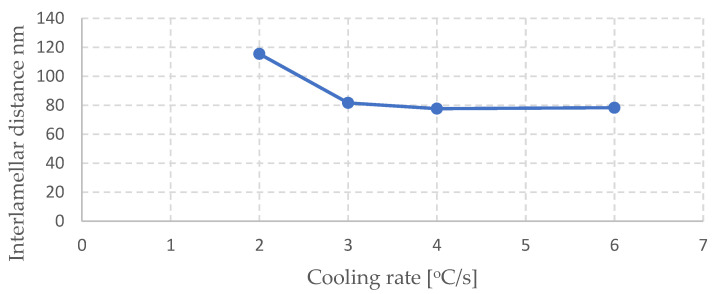
Relation between the interlamellar distance and cooling rate.

**Figure 12 materials-16-06430-f012:**
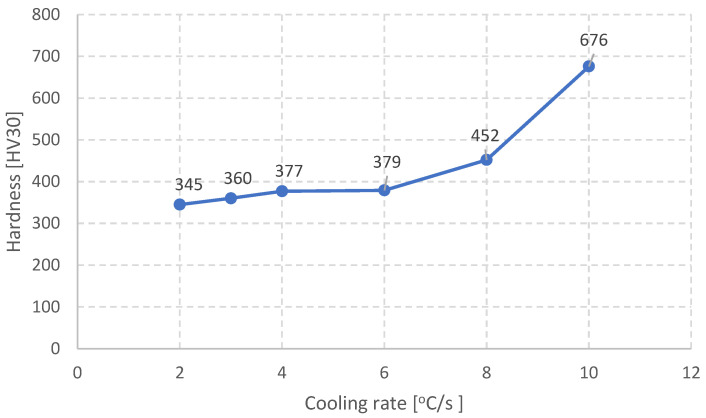
The relation between the cooling rate and the steel hardness.

**Figure 13 materials-16-06430-f013:**
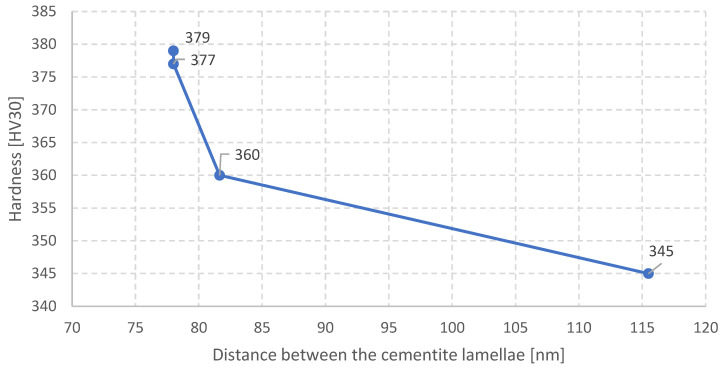
Relation of the distance between the cementite lamellae and the hardness.

**Figure 14 materials-16-06430-f014:**
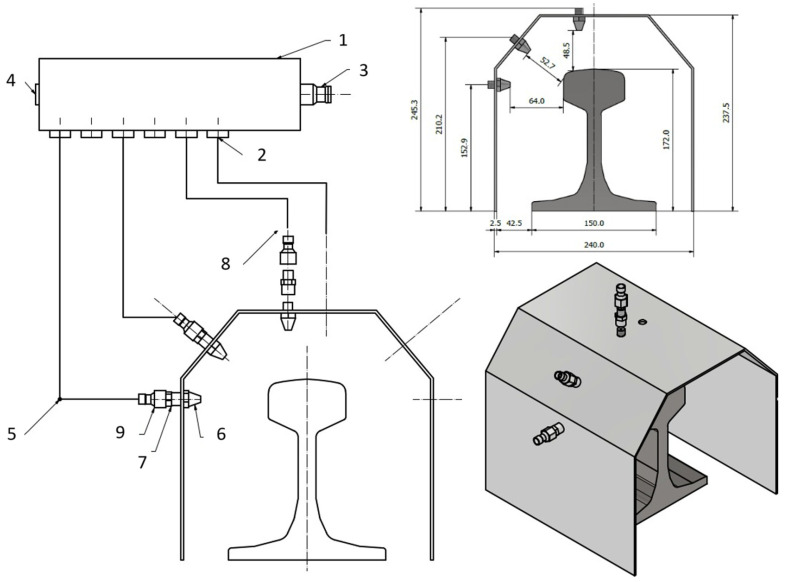
Diagram of the station used to cool samples made of railway rails: 1—connecting strip, 2—push fit, 3—connection ES 14 NA, 4—plug, 5—hydraulic hose, ϕ = 6 mm (used in the whole construction), 6—diffuser nozzle, 7—jointing sleeve, 8—throttle valve, 9—push fit.

**Figure 15 materials-16-06430-f015:**
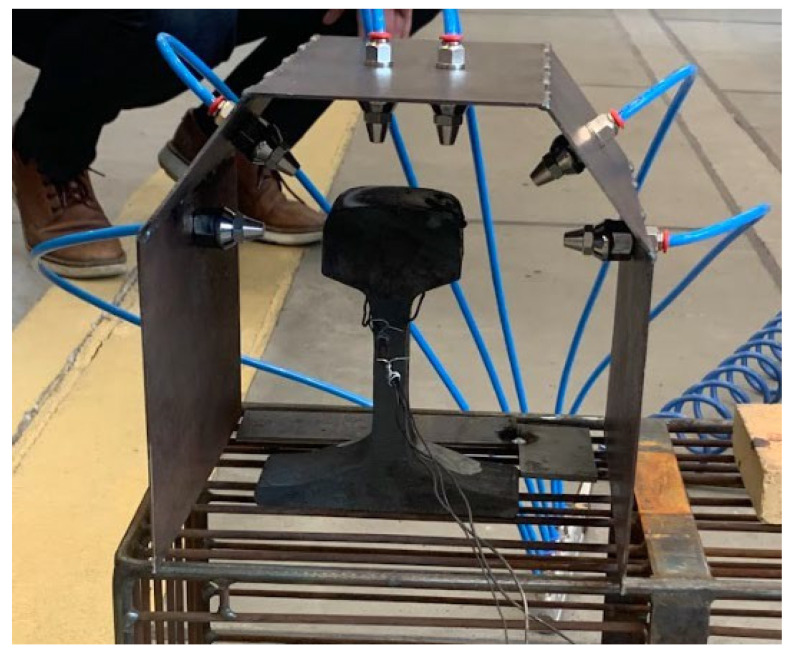
View of the real testing station (own design).

**Figure 16 materials-16-06430-f016:**
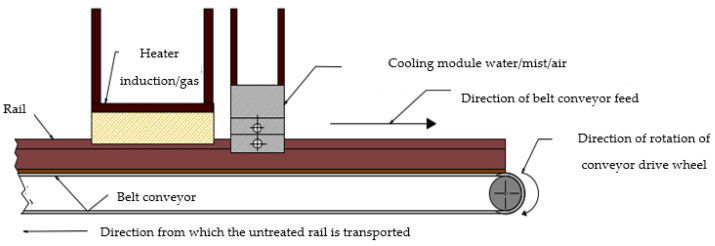
Concept of an industrial station for thermal treatment of rails.

**Figure 17 materials-16-06430-f017:**
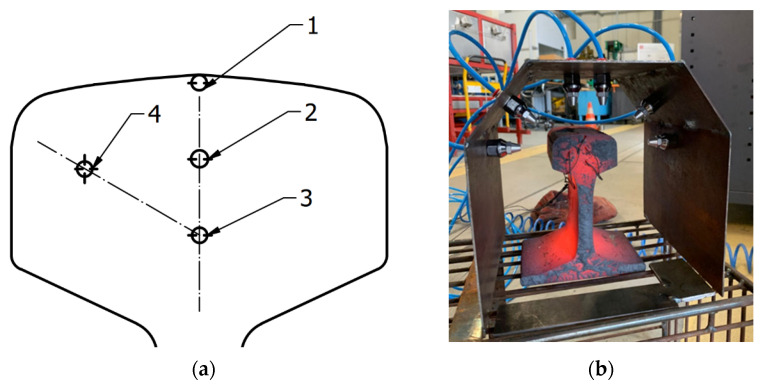
Technique of temperature measurement: (**a**) cross-section of the rails with the marked measurement points and (**b**) photograph of the rail with the connected thermocouples (the temperatures were measured in 1–4 points).

**Figure 18 materials-16-06430-f018:**
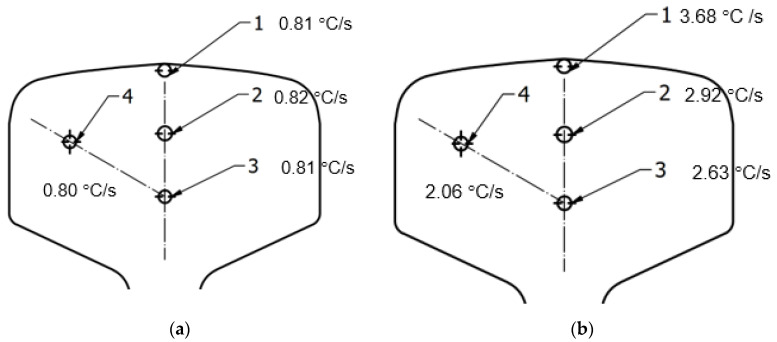
Cooling rate for the pressures (**a**) 0.5 bar and (**b**) 6.5 bar.

**Figure 19 materials-16-06430-f019:**
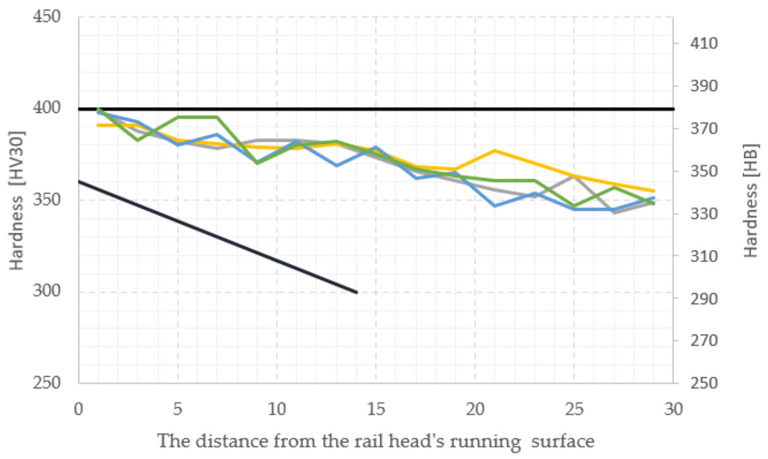
Results from semi-industrial tests of hardness in the required region according to the standard EN 16273 along the line from T1 to T4 (Figure 1a) [5].

**Figure 20 materials-16-06430-f020:**
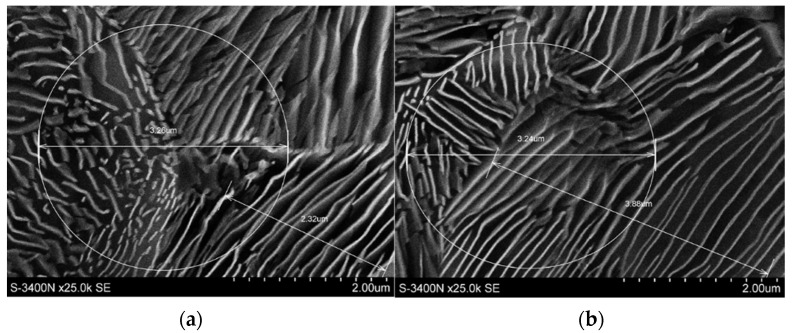
Microstructure of the pearlite in a rail after cooling on the stand at a rate of (**a**) 3.68 °C/s and (**b**) 2.63 °C/s.

**Table 1 materials-16-06430-t001:** Chemical composition of eutectoidal steels used to produce railway rails [5,6].

Steel Grade	Chemical Composition, Mass %						R_m_	A	Hardness
	C	Si	Mn	P	S	V	(MPa)	(%)	HB
R260	0.62–0.8	0.13–0.60	0.65–1.25	≤0.030	≤0.030	0.03	880	10	260–300
R350HT	0.72–0.8	0.13–0.60	0.65–1.25	≤0.025	≤0.030	0.01	1175	9	350–900
U75V	0.71–0.8	0.5–0.8	0.95–1.05	≤0.025	≤0.025	0.04–0.12	≥980	≥10%	280–320

**Table 2 materials-16-06430-t002:** Determined temperatures of the transformation’s beginning and end.

Cooling Rate	2 (°C/s)	3 (°C/s)	4 (°C/s)	6 (°C/s)	8 (°C/s)	10 (°C/s)
Transformation start temperature	647 °C	620 °C	611 °C	594 °C	Unnoticeable	Unnoticeable
Transformation finish temperature	564 °C	548 °C	521 °C	516 °C	Unnoticeable	Unnoticeable

**Table 3 materials-16-06430-t003:** Results of the measurement of the interlamellar distance for a cooling rate of 2 °C/s.

No of Sample	d (μm)	O=L	N1	N2	N3	S (nm)
°C/s						
1	3.2	10.0531	13	16	12	122.6
2	3.3	10.3673	14	16	10	129.6
3	3.3	10.3673	12	15	17	117.8
4	3.1	9.73894	18	17	13	101.4
5	3.24	10.1788	17	16	16	103.9

**Table 4 materials-16-06430-t004:** Cooling rates for all the applied pressures.

Pressure	Heating Temperature	Measurement Points			
		1	2	3	4
0.5 bar	900–950	0.81	0.82	0.81	0.8
1 bar	900–950	0.76	0.82	0.82	0.83
2 bar	900–950	1.56	1.43	1.40	1.3
3 bar	900–950	1.60	1.53	1.44	1.35
6.5 bar	900–950	3.68	2.92	2.63	2.06

## Data Availability

The data presented in this study are available on request from the corresponding author.

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
