# Peer review of "Advanced Heat Treatment of Pearlitic Rail Steel"

_materials, 2023, doi:10.3390/ma16196430_

Round 1

Reviewer 1 Report

General comments:

Overall, the authors' article is of high quality, relevant to mechanical engineering and/or material sciences, and clear and unambiguous in its objectives, results, and methods for achieving them. The obtained results, their processing, and the conclusions drawn are clear and appropriately presented.

The goal of this study is to conduct systematic investigations into the heat treatment of 60E1 profile rails made of steel R350HT to ensure the properties required by EN 16273. Furthermore, it will present a concept of cooling rails on a semi-industrial station that will allow the desired properties to be obtained. 

When both the EN 16273 standard's hardness distribution and microstructure requirements are met, dilatometric tests show that the optimal cooling rate is within the range of 3 °C/s to 6 °C/s. The required hardness will be reached only a few millimeters from the material's external surface for cooling rates of 2 °C/s, and for cooling rates of 8 °C/s and 10 °C/s, the requirements are not met, not only due to too high hardness, but also the presence of a bainitic structure, which is undesirable according to the standards, as well as degenerate pearlite.

Based on the results, we can conclude that increasing the cooling rate from 2 °C/s to 6 °C/s increases the material's hardness and decreases the distance between the cementite lamellae in the pearlite (Sp).

The study also developed a concept and design for a cooling station, which will allow for controlled cooling of an entire rail.

The interlamellar distance was also measured for these parameters: the cooling rate obtained at the surface was 3.68 °C/s, with an interlamellar distance of about 80 nm, whereas inside the rail, 2.63 °C/s and 110 nm.

The results show that the designed station can be used for controlled cooling tests on rail steels.

The reviewer requests the below modifications in the manuscript:

1. Decimal dots must be applied instead of decimal commas. Please double-check the entire paper and make the needed fixations!

2. The figures inserted in the article have poor resolution. This is true of all the charts (diagrams). In addition, a demanding author uses the same font types for all objects (graphs, tables, etc.). The reviewer asks that these be uninformed! Palatino Linotype is the recommended font type. (First of all, see the diagrams you prepared, the photos seem to be appropriate, however, where you put text in photos and drawings, the comment is also important). Please focus on formatting, i.e., font sizes (to be able to read the details, etc.), bold or normal font types, etc.

3. The notations are recommended to be formatted in italic font types. This comment is related to equations and the explanation of the notations in the text, and the appearance of these notations in the manuscript, everywhere.

4. It is recommended to give the hardness values everywhere in the paper not in HV30 unit, but in HB units. If the authors insist on the use of HV30, the reviewer requests that an HB scale be given in parallel in the diagrams as a supplement, or, if the unit HV30 is used in text or tables with values, that the value in HB be given in brackets in each occurrence.

5. Fig. 20(a) is not a figure; it must be formatted as a table.

6. Row #196: "R250HT", please revise!

The reviewer recommends the application and incorporation of the following relevant and up-to-date references:

1.

European Committee for Standardization, 2011, EN 13674-1:2011, Railway applications. Track. Rail. Part 1: Vignole railway rails 46 kg/m and above, 123 p.

2.

Fischer, S., Harangozó, D., Németh, D., Kocsis, B., Sysyn, M., Kurhan, D., Brautigam, A., 2023, Investigation of heat-affected zones of thermite rail weldings, Facta Universitatis Series Mechanical Engineering, doi: https://doi.org/10.22190/FUME221217008F.

3.

Barna, V., Brautigam, A., Kocsis, B., Harangozó, D., Fischer, S., 2022, Investigation of the Effects of Thermit Welding on the Mechanical Properties of the Rails, Acta Polytechnica Hungarica, 19(3), pp. 37-49. https://doi.org/10.12700/APH.19.3.2022.3.4

Reasons of the requests: Item #1 is the relevant European Standard related to railway rails. Items #2 and #3 are specialized journal papers which also investigated not only the micro- and macrostructures of the rails but also the variations in hardnesses due to rail welding that also uses additional heat and modifies the characteristics of the rail steel. In item #2, the authors used the R350HT and R400HT rail categories. The using of the above-mentioned papers and literature can improve the quality of the current questionable publication.

Questions:

1. What is the reason for the application of rail steel category R350HT? Do the authors plan to use and consider R400HT and/or bainitic rails in the future?

2. What is the reason for not giving an approximate correlation function in Fig. 20(b), and why isn't it compared with the specimens without heat treatment?

3. How much is the real/calculated accuracy of the fulfilled laboratory measurements (only the resolution ranges of the devices/instruments are given in Section 3)? Please supplement the missing information!

Minor editing of English language required

Author Response

Thank you very much for your important suggestions. We improved manuscript according to your remarks. All changes are marked in the text by red colour.

Full answer with figures is in pfd. file

Reviewer 1

  1. Decimal dots must be applied instead of decimal commas. Please double-check the entire paper and make the needed fixations!

Answer.

It was changed in all the text

  1. The figures inserted in the article have poor resolution. This is true of all the charts (diagrams). In addition, a demanding author uses the same font types for all objects (graphs, tables, etc.). The reviewer asks that these be uninformed! Palatino Linotype is the recommended font type. (First of all, see the diagrams you prepared, the photos seem to be appropriate, however, where you put text in photos and drawings, the comment is also important). Please focus on formatting, i.e., font sizes (to be able to read the details, etc.), bold or normal font types, etc.

Answer.

All figures were changed according to your suggestion

  1. The notations are recommended to be formatted in italic font types. This comment is related to equations and the explanation of the notations in the text, and the appearance of these notations in the manuscript, everywhere.

Answer.

It was changed in all the text

  1. It is recommended to give the hardness values everywhere in the paper not in HV30 unit, but in HB units. If the authors insist on the use of HV30, the reviewer requests that an HB scale be given in parallel in the diagrams as a supplement, or, if the unit HV30 is used in text or tables with values, that the value in HB be given in brackets in each occurrence.

Answer.

We added the HB scale

Fig. 1.

Fig. 20.

  1. 20(a) is not a figure; it must be formatted as a table.

Answer

We decided to present only figure because it is more legible, we removed table.

Fig. 20.

  1. Row #196: "R250HT", please revise!

Answer

It was changed

Questions:

  1. What is the reason for the application of rail steel category R350HT? Do the authors plan to use and consider R400HT and/or bainitic rails in the future?

Answer:

In the past, the most rails were made of R260 steel, recently R350HT steel rails have been introduced, but the heat treatment process is complicated in the case of large-scale production. Many metallurgical companies are trying to optimize this technology for final properties and performance, but the literature provides very little information on the heat treatment of R350HT steel. – it was added to the text.

Explanation for second part of the question, which was not added to the text.

We are going to research the bainitic rails, even we apply for project and we are negotiating with firm about it however due to hard time for industry it is difficult topic

  1. What is the reason for not giving an approximate correlation function in Fig. 20(b), and why isn't it compared with the specimens without heat treatment?

The most important was to fulfil requirements of the standard EN 16273 referring to the location of the hardness measurement and the scope of acceptable hardness depending on the distance from the rail head’s front surface.

Answer

Due to your suggestion we added sentences.

The hardness rail from R350HT steel without heat treatment equals to about 320 HV30 (300HB) in all the cross-section, which is much lower than requirements.

  1. How much is the real/calculated accuracy of the fulfilled laboratory measurements (only the resolution ranges of the devices/instruments are given in Section 3)? Please supplement the missing information!

Answer

We added following text

The cooling temperature was measured with TandD's MCR-4TC, where all thermocouples were mounted at the same time. Sampling rate was 1 Hz, data was collected on the course of the entire test (during heating and cooling). The cooling rate was determined by a graphical method on the basis of the curve from start of accelerated cooling to pearlitic transformation. Accuracy for thermocouple K was , ±(0.5 °C + 0.3 % of reading) and the operating range for the set was -270 to 1370 °C.

The reviewer recommends the application and incorporation of the following relevant and up-to-date references:

We added following references and text

This European Standard railway rails are given in two norms [5, 6].

Research is also carried out on the influence of welding parameters on the microstructure of pearlite. In the manuscript [22], the authors presents the influence of thermit welding on the mechanical properties of the rails. Developing this research is the analyse of microstructure which proves that after welding fine pearlite grains and large coarse pearlite grains are formed as a function of the cooling rate. [23].

[6] European Committee for Standardization, 2011, EN 13674-1:2011, Railway applications. Track. Rail. Part 1: Vignole railway rails 46 kg/m and above, 123 p.

[22] Barna, V., Brautigam, A., Kocsis, B., Harangozó, D., Fischer, S., 2022, Investigation of the Effects of Thermit Welding on the Mechanical Properties of the Rails, Acta Polytechnica Hungarica, 19(3), pp. 37-49. https://doi.org/10.12700/APH.19.3.2022.3.4

[23] Fischer, S., Harangozó, D., Németh, D., Kocsis, B., Sysyn, M., Kurhan, D., Brautigam, A., 2023, Investigation of heat-affected zones of thermite rail weldings, Facta Universitatis Series Mechanical Engineering, doi: https://doi.org/10.22190/FUME221217008F.

Reviewer 2 Report

1. "The research was conducted for two fully pearlitic structures, with the distances between the cementite lamellae of 170 and 230 nm, the yield point of 396 MPa and 498 MPa and the hardness of 220 HV50 and 270 HV50, was respectively obtained. "  This sentense maybe make a mistake. Please explain why the cementite lamellae was smaller but the yield point and the hardness was lower.

2."That is, the nozzles cooling the pitch surface were connected to opening 1. and 2., the rounded part of the head-to opening 3. and 4., and the pitch surface-to opening 5. and 6"  I can not find these six openings in the figure 14 or 16. Please show them in their position.

3. In Figure 19 (b) and Table 4 showed when cooling rate for the pressure of 6,5 bar, the point 4  had the lowest cooling rate 2.06°C/s. Please explain the reason and supply the obtained interlamellar distance of each point respectively. 

Author Response

Thank you very much for your important suggestions. We improved manuscript according to your remarks. All changes are marked in the text by red colour.

Full answer with figures is in pfd. file

Reviewer 2#

  1. "The research was conducted for two fully pearlitic structures, with the distances between the cementite lamellae of 170 and 230 nm, the yield point of 396 MPa and 498 MPa and the hardness of 220 HV50 and 270 HV50, was respectively obtained. "  This sentense maybe make a mistake. Please explain why the cementite lamellae was smaller but the yield point and the hardness was lower.

Answer

It was my mistake

It was changed into

The research was conducted for two fully pearlitic structures, with the distances be-tween the cementite lamellae of 230 and 170 nm, the yield point of 396 MPa and 498 MPa and the hardness of 220 HV50 and 270 HV50, was respectively obtained.

2."That is, the nozzles cooling the pitch surface were connected to opening 1. and 2., the rounded part of the head-to opening 3. and 4., and the pitch surface-to opening 5. and 6"  I can not find these six openings in the figure 14 or 16. Please show them in their position.

Answer. In the figure all 6 nozzles are marked

  1. In Figure 19 (b) and Table 4 showed when cooling rate for the pressure of 6,5 bar, the point 4  had the lowest cooling rate 2.06°C/s. Please explain the reason and supply the obtained interlamellar distance of each point respectively. 

Answer:

We added following text. The lowest cooling rate in the most experiments was at the point 4, however cooling rates at point 3 and 2 were very similar therefore interlamellar distance for 6,5 bara was determined only for point 1 and 3 and was about 80 nm and 110 nm, respectively

Reviewer 3 Report

Pearlite steels are widely used materials in railway engineering and not only there. Their systematic study and improvement of various characteristics and properties is extremely important from an engineering and technical point of view. The title should be refined so that it sounds and corresponds to the scientific research carried out.

1. It is necessary to rework the Introduction section in a way that clearly considers the literature review made on the topic.

2. All types of data, for example chemical composition of the investigated materials, are presented in the Results and Discussion section. It is not clear whether the results presented in Table 1 are from own research or are they taken from literature data?

3. The methods are not described according to the requirements for the construction of a scientific article.

4. The presented figures are quite numerous, which makes it difficult to read and understand the scientific results. 

A little English language check is needed for grammatical and stylistic errors.

Author Response

Thank you very much for your important suggestions. We improved manuscript according to your remarks. All changes are marked in the text by red colour.

Full answer with figures is in pfd. file

Reviewer 3#

  1. It is necessary to rework the Introduction section in a way that clearly considers the literature review made on the topic.

Answer

 Introduction was rewrite. Additionally 5 references were added. We had to take into consideration also other reviews. All changes are marked in the text by red colour.

  1. All types of data, for example chemical composition of the investigated materials, are presented in the Results and Discussion section. It is not clear whether the results presented in Table 1 are from own research or are they taken from literature data?

Answer

In the table 1 there are chemical composition of the most popular rail steels taken form standards.

  1. The methods are not described according to the requirements for the construction of a scientific article.

Answer

Taking into consideration also other reviews we changed the text. All changes are marked in the text by red colour.

  1. The presented figures are quite numerous, which makes it difficult to read and understand the scientific results. 

Answer

We removed figures 14 and 20 a, also all figures were corrected and improved according to your and other reviewer remarks.   

Reviewer 4 Report

The authors investigate the temperature regime within the head of a railroad rail during a heat treatment designed to develop an optimum pearlitic structure.  The results of this exploration permit the engineering design and construction of a heat treatment unit for steel rails in alloy R350HT.

This reviewer finds merely minor issues which would benefit from the attention of the authors:

37. ‘low value of distance’ would more usually read ‘fine inter-lamellar spacing’

63. The equation requires to be clarified with appropriate use of brackets. For instance do the authors mean -0.52[exp(-SP/58.29) + 0.32] or -0.52exp[(-SP/58.29) + 0.32].  Please clarify.

Fig 11. Bar charts are not helpful in scientific and technical reports. This data should be presented as a graph of interlamellar spacing versus cooling rate. (as a graph the data can be interpolated and extrapolated easily, and is more intuitively understood by the reader).

Avoid the ÷ sign which is normally used to denote ‘division’.  Use the word ‘to’.

Fig 20. Do the black lines on the graph represent limits to the temperature regime? Please clarify.

398. Suggest the word ‘acceptably’ in place of ‘properly’.

424. Suggest delete the fractions of nanometers as impractically and unreasonably precise. The values would more reasonably read 77 to 115 nm.

To comment more generally on their work, it seems to this reviewer that the authors have achieved a novel heat treatment unit for rails for which they should be congratulated. In fact the invention may apply widely to other long steel alloy products, including wire and bar stock. 

However, they may wish to protect their invention by patent.  If so, they may wish this publication to be delayed until a provisional patent application is secured.  When this submission is published, thereby released into the public domain, as I hope it will be, the opportunity for seeking a patent will have been missed.  

If the authors have already applied for a patent, there is an obligation on the authors to declare this, stating ‘patent applied for’ or ‘Patent Number ………’.

A few minor suggestions have been listed.

Author Response

Thank you very much for your important suggestions. We improved manuscript according to your remarks. All changes are marked in the text by red colour. 

Full answer with figures is in pfd. file

Reviewer 4#

The authors investigate the temperature regime within the head of a railroad rail during a heat treatment designed to develop an optimum pearlitic structure.  The results of this exploration permit the engineering design and construction of a heat treatment unit for steel rails in alloy R350HT.

This reviewer finds merely minor issues which would benefit from the attention of the authors:

-37. ‘low value of distance’ would more usually read ‘fine inter-lamellar spacing’

Answer

It was changed

-63. The equation requires to be clarified with appropriate use of brackets. For instance do the authors mean -0.52[exp(-SP/58.29) + 0.32] or -0.52exp[(-SP/58.29) + 0.32].  Please clarify.

Answer

It was changed into

-Fig 11. Bar charts are not helpful in scientific and technical reports. This data should be presented as a graph of interlamellar spacing versus cooling rate. (as a graph the data can be interpolated and extrapolated easily, and is more intuitively understood by the reader).

Avoid the ÷ sign which is normally used to denote ‘division’.  Use the word ‘to’.

Answer.

We change figure 11.

Figure 11

-Fig 20. Do the black lines on the graph represent limits to the temperature regime? Please clarify.

We added explanation

Figure 20. Results from semi-industrial tests of hardness in the required region according to the standard EN 16273 along the line from T1 to T4 (Figure 1a) [5].

-398. Suggest the word ‘acceptably’ in place of ‘properly’.

Answer

It was changed

-424. Suggest delete the fractions of nanometers as impractically and unreasonably precise. The values would more reasonably read 77 to 115 nm.

Answer

It was changed

To comment more generally on their work, it seems to this reviewer that the authors have achieved a novel heat treatment unit for rails for which they should be congratulated. In fact the invention may apply widely to other long steel alloy products, including wire and bar stock. 

However, they may wish to protect their invention by patent.  If so, they may wish this publication to be delayed until a provisional patent application is secured.  When this submission is published, thereby released into the public domain, as I hope it will be, the opportunity for seeking a patent will have been missed. 

Answer

Thank you very much for your valuable suggestion

We have prepared the description of patent

The presented cooling station has been submitted for patent protection.

Round 2

Reviewer 1 Report

Dear Authors,

many thanks for the revised version. I recommend the paper for publishing.

Yours sincerely,

The Reviewer.